# The Cell-Penetrating Peptide GV1001 Enhances Bone Formation via Pin1-Mediated Augmentation of Runx2 and Osterix Stability

**DOI:** 10.3390/biom14070812

**Published:** 2024-07-08

**Authors:** Meiyu Piao, Sung Ho Lee, Jin Wook Hwang, Hyung Sik Kim, Youn Ho Han, Kwang Youl Lee

**Affiliations:** 1Research Institute of Pharmaceutical Sciences, College of Pharmacy, Chonnam National University, Gwangju 61186, Republic of Korea; my101park@gmail.com (M.P.); puzim23@jnu.ac.kr (S.H.L.); 2INSERM UA09, University Paris Saclay, 94800 Villejuif, France; jinwook.hwang@inserm.fr; 3School of Pharmacy, Sungkyunkwan University, 2066, Seobu-ro, Jangan-gu, Suwon 16419, Republic of Korea; hkims@skku.edu; 4Department of Oral Pharmacology, College of Dentistry, Wonkwang University, Iksan 54538, Republic of Korea

**Keywords:** GV1001, biotin-conjugated GV1001, osteoblast differentiation, Pin1

## Abstract

Peptide-based drug development is a promising direction due to its excellent biological activity, minimal immunogenicity, high in vivo stability, and efficient tissue penetrability. GV1001, an amphiphilic peptide, has proven effective as an anti-cancer vaccine, but its effect on osteoblast differentiation is unknown. To identify proteins interacting with GV1001, biotin-conjugated GV1001 was constructed and confirmed by mass spectrometry. Proteomic analyses were performed to determine GV1001’s interaction with osteogenic proteins. GV1001 was highly associated with peptidyl-prolyl isomerase A and co-immunoprecipitation assays revealed that GV1001 bound to peptidyl-prolyl cis-trans isomerase 1 (Pin1). GV1001 significantly increased alkaline phosphatase (ALP) activity, bone nodule formation, and the expression of osteogenic gene markers. GV1001-induced osteogenic activity was enhanced by *Pin1* overexpression and abolished by *Pin1* knockdown. GV1001 increased the protein stability and transcriptional activity of Runx2 and Osterix. Importantly, GV1001 administration enhanced bone mass density in the OVX mouse model, as verified by µCT analysis. GV1001 demonstrated protective effects against bone loss in OVX mice by upregulating osteogenic differentiation via the Pin1-mediated protein stabilization of Runx2 and Osterix. GV1001 could be a potential candidate with anabolic effects for the prevention and treatment of osteoporosis.

## 1. Introduction

Osteoporosis is a metabolic bone disease with symptoms such as low bone density, bone structure deterioration, and increased risk of fractures. Fractures associated with osteoporosis can lead to increased pain, disability, and mortality, along with high medical costs [1]. Typical osteoporosis treatments include bisphosphonate, estrogen agonists/antagonists, parathyroid hormone analog, calcitonin, and recently, developed drugs using the new mechanism, including monoclonal antibodies, against sclerostin and cathepsin K inhibitors [2]. Common side effects of bisphosphonates, the most commonly prescribed drugs, are stomach disorders with symptoms such as heartburn, and the most aggressive side effects are osteonecrosis of the jaw and atypical femur fractures [3]. Other recently developed drugs, including monoclonal antibodies, have not yet been approved for long-term use [4]. Therefore, it is necessary to develop drugs that demonstrate potency with stability. With respect to that, peptide drugs with high safety margins are likely to be alternative new osteoporosis treatments if proven effective. The formation of new bone tissue depends largely on the function of osteoblasts [5]. A decrease in the functional capacity of osteoblasts is one of the major causes of osteoblast loss. In old bones, the replenishment of old osteoblasts does not meet the requirements for bone remodeling [6]. Thus, understanding the mechanism underlying the regulation of bone regeneration is fundamental for the prevention and treatment of osteoporosis.

Peptides are naturally occurring biological molecules that perform a wide range of essential functions in all living organisms and function as biological transmitters of information from one cellular component or system to another [7]. As essential signaling molecules in various physiological functions, peptides exert a therapeutic function via natural pathways [8]. Clinically, several peptide drugs are used in hormone replacement therapy to supplement essentially deficient or absent endogenous levels of hormones [9]. Along with insulin, more than 60 peptide drugs have been approved in major drug markets, and peptides continue to advance into clinical development at a steady pace [10]. The early development of peptide drugs focused primarily on using endogenous human peptides but gradually diversified to include various structures identified through different natural sources or medicinal chemical activities [10,11].

The peptide used in this study, GV1001, is a short peptide with 16 amino acids (Figure 1A) [12]. GV1001 is an anti-cancer drug that inhibits cell viability and induces apoptosis in castration-resistant prostate cancer cells through the AKT/NF-κB/VEGF pathway [13]. Recently, it has been reported that GV1001 is a novel ligand for the gonadotropin-releasing hormone receptor (GnRHR) by selectively stimulating the Gαs/cAMP pathway, suggesting the possibility of a treatment for Alzheimer’s disease as well as cancer [14]. The activation of the GnRHR would, in turn, induce an increase in the synthesis of estrogen in the ovary by the hypothalamic–pituitary–gonadal axis and based on estrogen’s association with osteoporosis [15,16], we initiated a study on the potential of GV1001 as a treatment for osteoporosis. Furthermore, GV1001 is not positively charged at physiological pH and demonstrates amphiphilic (water and lipid-loving) characteristics. Therefore, it exhibits a cell-penetrating capacity, allowing the transport of different types of cargos into cells [17]. Given that bones are more difficult to penetrate than other tissues for drugs [18], this feature of GV1001 can work predominantly in the determination of drug efficacy. However, to fully understand how GV1001 exerts its effects, identifying specific molecular targets within bone cells is essential. We addressed this limitation by developing a biotinylated derivative, biotinylated GV1001. Biotin, a small vitamin with a remarkable affinity for avidin and streptavidin proteins, allows for conjugation to various proteins without altering their biological activities and enables detection in assays, like proteomics analysis, using streptavidin probes.

In the present study, we first constructed biotinylated GV1001 to identify the molecular targets during the osteoblast differentiation and further investigated whether the GV1001 peptide could regulate osteoblast differentiation and bone formation in vitro and in vivo.

## 2. Materials and Methods

### 2.1. Plasmid, Antibodies, and Reagents

Plasmids for Myc-Runx2 and Myc-Osterix were developed in a CMV promoter-containing expression vector (pCS4+). The peptidyl-prolyl cis-trans isomerase 1 (Pin1) plasmid was kindly provided by Dr. Choi (Chosun University, Korea). Anti-Runx2 (#8486), anti-α-Tubulin (#3873) and anti-Myc (#9402) antibodies were purchased from Cell Signaling Technology (Beverly, MA, USA). Short hairpin RNAs (shRNAs) containing a mouse *Pin1* silencing sequence (forward: 5′-GAT CCC CGC CGG GTG TAC TTC AAT TCA AGA GAT TGA AGT ACA CCC GGC TTT TTG GAA A-3′; reverse: 5′-AGC TTT TCC AAA AAG CCG GGT GTA CTA CTT CAA TCT CTT GAA TTG AAG TAG TAC ACC CGG CGG G-3′) were used to mediate Pin1 knockdown. Anti-Osterix (sc-393325) was purchased from Santa Cruz Biotechnology (Dallas, TX, USA). Recombinant mouse BMP4 protein was purchased from the R&D system (314-BP, Minneapolis, MN, USA) and cycloheximide (CHX) was obtained from Sigma-Aldrich (66-81-9, St. Louis, MO, USA). GV1001 was obtained from AnyGen (Gwangju, Republic of Korea).

### 2.2. Cell Culture and Transient Transfection

The pre-myoblast C2C12 cell line and *Pin1−/−* mouse embryonic fibroblast (MEF) cell line were cultured in Dulbecco’s modified Eagle’s medium (DMEM; Gibco BRL, Grand Island, NY, USA) supplemented with 10% fetal bovine serum (FBS; Gibco BRL) and 1% penicillin-streptomycin antibiotics (Thermo Fisher Scientific, Waltham, MA, USA) at 37 °C in an incubator containing 5% CO_2_. Polyethylenimine (PEI; Polysciences, Warrington, PA, USA) was used for transient transfection. Cells were plated in 24-well plates 1 d before transfection at a density of 2 × 10^4^ cells/well. After exchanging the medium, the cells were co-transfected with the aforementioned plasmids.

### 2.3. Biotin Conjugation Reaction

In the biotin conjugation reaction, the GV1001 conjugates were synthesized using the N-Hydroxysuccinimide ester (NHS) method. Then, biotinylated GV1001 was incubated with the labeled streptavidin conjugate for 2 h in the dark. After incubation, all of the reactions were deposited by centrifugation and washed with the sodium bicarbonate buffer (pH 9.6, 25 mM) to remove excess unreacted biotin. In total, biotinylated GV1001 was obtained as a white solid. The structure was supported by TOF mass spectrometry.

### 2.4. Determination of GV1001-Binding Proteins by Matrix-Assisted Laser Desorption Ionization Mass Spectrometry (MALDI-MS)

C2C12 cells were cultured until reaching 80% confluence, following which they were exposed overnight to either GV1001 or biotinylated GV1001. Post-treatment, cells were rinsed with PBS and lysed using a RIPA buffer. The resultant cell lysate was diluted at a ratio of at least 1:10 with a urea buffer and combined with 50 µL of streptavidin resin. This mixture was gently agitated at 4 °C for 1 h. After a brief centrifugation, the resin was washed twice with a urea buffer and then resuspended in 50 µL of a 1× SDS loading buffer for subsequent separation via polyacrylamide gel electrophoresis. Both the non-GV1001-binding fraction (flowthrough) and the GV1001-binding fraction (elution) were loaded onto the gel for separation.

Following electrophoresis, the gel was stained with Coomassie blue to visualize the proteins. Identification of proteins captured by streptavidin was achieved by comparing the elution fraction of GV1001-treated cells with that of GV1001 biotin-treated cells. Protein bands of interest were excised from the gel, subjected to in-gel tryptic digestion, and analyzed using MALDI-MS as previously described [19]. Protein identification was performed using the MASCOT Peptide Mass Fingerprint online search engine “https://www.matrixscience.com” (accessed on 1 July 2024). Each identified protein was assigned a probability-based Mowse score, and the number of matched peptides was determined. A Mowse score exceeding 45 suggests identity or significant homology, with higher scores and a greater number of matched peptides, indicating a higher likelihood of genuine binding events.

### 2.5. Pin1 Activity Assay

Pin 1 activity levels were assessed utilizing the SensoLyte^®^ Green Pin 1 activity assay kit (AnaSpec, Fremont, CA, USA), following the manufacturer’s guidelines. In summary, a Pin 1 substrate solution was combined with the indicated combination and allowed to incubate for 1 h at room temperature. Subsequently, the fluorescence signal was measured using a Multi-Mode Microplate Reader System (Perkin-Elmer, Waltham, MA, USA) with excitation and emission wavelengths set at 490 nm and 520 nm, respectively.

### 2.6. Alkaline Phosphatase (ALP) and Alizarin Red S (ARS) Staining

C2C12 cells (3 × 10^4^ cells/well) were cultured in 24-well plates and incubated with BMP4 (30 ng/mL) in combination with the indicated concentration of GV1001 for 3 days (for ALP staining) or 7 days (for ARS staining). After fixing the cells in 4% formaldehyde for 5 min, ALP activation was evaluated using the color of the BCIP/NBT substrate (Sigma-Aldrich, St. Louis, MO, USA). The cells were stained with 2% Alizarin Red S (Sigma-Aldrich) to observe bone mineralization. After washing with PBS, the absorbance of the samples dissolved in DMSO was measured at 610 nm (ALP staining) or 405 nm (ARS staining) using a SpectraMax 190 microplate reader (Molecular Devices, San Jose, CA, USA).

### 2.7. RNA Extraction and Real-Time Reverse Transcription Polymerase Chain Reaction (RT-PCR)

Total RNA was extracted using the Trizol reagent (Thermo Fisher Scientific, Waltham, MA, USA) according to the manufacturer’s instructions. For cDNA synthesis, a random primer and reverse transcriptase (Toyobo, Osaka, Japan) were added to 1 μg of total RNA. Real-time PCR was carried out using the Thunderbird SYBR qPCR Master Mix (Toyobo) and StepOne Real-Time PCR System (Molecular Devices, San Jose, CA, USA). The procedure involved an initial reaction step at 95 °C for 30 s, followed by 40 cycles comprising 5 s denaturation at 95 °C and 30 s annealing at 60 °C to derive the Ct values, ensuring uniqueness across all analyses. The expression level of the housekeeping gene *Gapdh* was used as a reference to normalize the variability and was analyzed using the 2^−ΔΔCt^ method. The primer sequences are listed in Table 1. The specificity of the primer sequences for each gene was confirmed by NCBI Primer-BLAST software “https://www.ncbi.nlm.nih.gov/tools/primer-blast/index.cgi” (accessed on 2 July 2024).

### 2.8. Immunoblotting (IB) and Co-Immunoprecipitation (Co-IP)

C2C12 cells were harvested and lysed with a RIPA lysis buffer (Thermo Fisher Scientific). After centrifugation (12,000× *g* at 4 °C for 15 min), the supernatants containing the protein (protein samples) were collected. A BCA protein assay kit (Thermo Fisher Scientific) was used to measure the protein concentration. For IB, the protein lysate (40 µg) was isolated using sodium dodecyl sulfate–polyacrylamide gel electrophoresis (SDS-PAGE) and transferred to polyvinylidene difluoride membranes (PVDF; GE Healthcare, Buckinghamshire, UK). The membranes were blocked with 5% non-fat milk and incubated overnight with primary antibodies at 4 °C. Subsequently, the membranes were washed with a TBS-T buffer and incubated with the appropriate horseradish peroxidase-conjugated secondary antibody (Thermo Fisher Scientific). Blots were visualized using an enhanced chemiluminescence kit (GE Healthcare). The band strength was quantified using the image software Multi Gauge V3.0 (FUJIFILM, Tokyo, Japan). For co-IP, HEK 293 cells were transiently transfected, indicating a combination of plasmids. Subsequently, the centrifuged lysate supernatants underwent IP with specific antibodies and protein A Sepharose beads. The immunoprecipitated proteins were separated by SDS-PAGE and visualized via IB. Original Western Blotting figures can be found in Appendix A. 

### 2.9. Luciferase Reporter Assay

C2C12 cells were co-transfected with plasmids, encoding ALP-Luc (luciferase), bone sialoprotein (BSP)-Luc, or an osteocalcin (OCN)-Luc reporter, and β-galactosidase (β-gal) was used as a control to analyze the transfection efficiency. After transfection, cells were treated with the indicated concentration of GV1001. After 24 h, cells were harvested in a reporter lysis buffer (Promega, Madison, WI, USA). Luciferase activity was determined in whole-cell lysates using the Promega luciferase assay kit.

### 2.10. Enzyme-Linked Immunosorbent Assay (ELISA) for OCN Detection

The secreted OCN protein was detected using the osteocalcin ELISA kit (Biomedical Technologies, Stoughton, MA, USA) according to the manufacturer’s instructions. For the analysis of extracellular OCN, 100 μL of the cell culture medium that was collected at different points during osteogenic differentiation was used.

### 2.11. Animal Models and Treatments

Female-specific pathogen-free (SPF) mice (7 weeks old) were purchased from Samtako Bio Korea. The mice were acclimatized for at least 2 weeks before initiating the experiments under controlled temperature (21 ± 2 °C) and humidity (50 ± 5%) and exposed to a 12 h light/dark cycle with ad libitum access to food and water. The mice were anesthetized with tribromoethanol (Avertin) and bilaterally ovariectomized (OVX). Mice belonging to the control group were sham operated for comparison. After 4 weeks of recovery following surgery, the OVX mice were randomly divided into four subgroups based on the administered treatment: (1) the OVX control group, (2) OVX mice administrated with GV1001 (0.1 mg/kg), (3) OVX mice administrated with GV1001 (1 mg/kg), and (4) OVX mice administered with GV1001 (10 mg/kg). GV1001 in saline was administered to GV1001-treated groups via a subcutaneous injection 5 days per week for 21 days; the SHAM and OVX control mice were administered the same volume of saline. At the end of the injection period, the mice were euthanized, and the bilateral femurs were isolated and collected. The mice femurs were fixed in 4% paraformaldehyde for further experiments. The animal handling and experimental procedures were performed in accordance with the regulations and rules of the Animal Ethics Committee of Sungkyunkwan University (SKKUIACUC2018-08-07-1).

### 2.12. Micro-Computed Tomography (µCT)

For the analysis of bone morphometric parameters and microarchitectural properties, the left-side femurs of the mice were scanned with a cone beam µCT (Skyscan 1172, Kontich, Belgium) according to a previously published protocol [20]. A total of 200 slices with a voxel size of 15 µm were directly analyzed above the growth plate of the distal femur. Microstructural parameters of the trabecular bone, including bone mineral density (BMD), bone volume/tissue volume (BV/TV), trabecular thickness (Tb.Th), trabecular number (Tb.N), structure model index (SMI), trabecular pattern factor (Tb.Pf), and trabecular space (Tb.Sp) were determined.

### 2.13. Statistical Analysis

All results are expressed as mean ± SD. Statistical comparisons between the two groups were assessed using Student’s *t*-test. Differences among multiple groups were evaluated using one-way analysis of variance (ANOVA) with the Tukey–Kramer test using GraphPad Prism 5.03 software (La Jolla, CA, USA). *p* < 0.05 was considered statistically significant.

## 3. Results

### 3.1. Biotinylated GV1001 Associates with Peptidyl-Prolyl Isomerase A (PPIA) and GV1001 Regulates Pin1 Activity by Direct Interaction

Biotin and streptavidin are often used in affinity-based techniques, such as proteomics, to study and identify target molecules and protein–protein interactions associated with various biological processes and diseases [21]. Therefore, we constructed biotinylated GV1001 to discover a protein capable of interacting with GV1001 (Figure 1B). As shown in Figure 1C, the condensation of GV1001 and N-Boc-2,2′-(ethylenedioxy) diethylamine using coupling reagents 1-ethyl-3-(3-dimethylaminopropyl) carbodiimide (EDC)–hydroxybenzotriazole (HOBt) and N, N-Diisopropylethylamine (DIPEA) in dimethylformamide (DMF) yielded the desired intermediate 1. Then, sequential deprotection of N-Boc in intermediate 1 using methylene chloride (MC): trifluoroacetic acid (TFA) (1:1) afforded the desired intermediate 2. Finally, the coupling of derivative 2 and biotin successfully gave biotinylated GV1001. The GV1001 peptide, when coupled with N-Boc-2,2′-(ethylenedioxy)diethylamine, exhibits an increase of 248 Da in molecular weight compared to the original molecule. Subsequently, through a series of de-protections of Boc (−100 Da) and biotinylation (+244 Da) processes, the biotinylated GV1001 gains an additional molecular weight of 392 Da. The observed increase in molecular weight can be confirmed through MS/MS spectrum analysis, where shifts are evident (Figure 1D). To compare the protein expression levels in each sample, MALDI-MS analysis and the MASCOT Peptide Mass Fingerprint online search engine were utilized. A list of proteins showing significant changes between protein samples extracted using streptavidin with GV1001 alone and biotinylated GV1001 was generated to screen proteins directly binding to the GV1001 peptide compound by protein identification analysis. Notable observed proteins in the biotinylated GV1001 sample were shown (Figure 1E). Interestingly, we identified that GV1001 was highly associated with PPIA, which is an enzyme that catalyzes the cis-trans isomerization of proline peptide bonds in proteins and influences bone metabolism, participating in both bone formation and resorption processes [22]. Specifically, Pin1, which is a major PPIA, modulates the phosphorylation status of multiple proteins in the BMP signaling pathway, thereby regulating crucial steps in osteogenesis [23]. The results led us to hypothesize that GV1001 modulates Pin1 activity through physical interaction. Co-immunoprecipitation (co-IP) assays confirmed that GV1001 is bound to Pin1 (Figure 1F). We next investigated whether GV1001 influences the catalytic activity of Pin1 using the Green Pin1 activity assay. In the group transfected with Pin1, the activity of Pin1 increased over time compared to the non-transfected control group (Figure 1G). Additionally, we also observed that the pure peptide of GV1001 significantly enhanced Pin 1 catalytic activity in comparison to the Pin1-transfected control, whereas the group treated with tannic acid, known to inhibit the activity of Pin1 used as a negative control, exhibited a suppression of Pin1 activity (Figure 1H). These data confirmed that biotinylated GV1001 was synthesized and found to interact with PPIA, specifically influencing its activity on Pin1, thereby suggesting a potential role for GV1001 in modulating Pin1 activity, as confirmed by the co-IP and Green Pin1 activity assay.

### 3.2. GV1001 Promotes Osteoblast Differentiation after Stimulation of C2C12 Cells with BMP4

Before investigating the influence of GV1001 on osteoblast differentiation through its relationship with Pin1, we first assessed the effect of GV1001 on osteoblast differentiation using the pre-myoblast cell line C2C12 that could differentiate osteoblasts upon the induction of osteogenesis by bone morphogenetic protein 4 (BMP4) with or without GV1001. GV1001 facilitated the initial stages of osteoblast differentiation by increasing the ALP activity and enhanced the terminal stages of osteoblast differentiation by increasing mineralization, as determined using ALP and Alizarin Red S (ARS) staining, respectively (Figure 2A). Next, the expression levels of the genes associated with osteoblast differentiation were investigated. The mRNA expression of *Alp*, an enzyme that is initially expressed during osteoblast differentiation, was increased upon BMP4 stimulation and further increased with GV1001 treatment. This pattern was similar to that of *Bsp*, which is known as a relatively medium-term marker. Although *Collα1* was not concentration-dependent on GV1001, it was significantly increased compared to the BMP-4 group when treated with the BMP-4 and GV1001 combination (Figure 2B). Next, the effect of GV1001 on the protein abundance of Runx2 and Osterix, known as the most important transcription factors in osteoblast differentiation, was examined. GV1001 increased the protein levels corresponding to Runx2 and Osterix at concentrations higher than 1 μg/mL in a dose-dependent manner (Figure 2C). In addition, the results of the experiment luciferase assay performed using the new ALP and BSP promoters to determine the effect on transcription activity during osteoblast differentiation showed that GV1001 increased the transcription activity after BMP4 stimulation in a concentration-dependent manner (Figure 2D). Those data indicate that GV1001 upregulates BMP4-induced osteoblast differentiation in C2C12 cells.

### 3.3. Knockdown of Pin1 Partially Reduced the Osteogenic Activity of GV1001

To determine whether the action of GV1001 on osteoblast differentiation is mediated by Pin1, the correlation of GV1001 with Pin1 knockdown was investigated. ALP staining showed that the effect of GV1001 on ALP activity was markedly attenuated by *Pin1* knockdown (Figure 3A). In addition, mRNA expression and protein levels of various osteogenic genes were enhanced by GV1001. However, the knockdown of *Pin1* abolished the osteogenic effects of GV1001 (Figure 3B,C). Next, the silencing of *Pin1* significantly reversed the enhanced transcriptional effect of GV1001 on the ALP and BSP promoters (Figure 3D). Moreover, the transcriptional effect of GV1001 was not observed in MEF cells with *Pin1* knockout; however, *Pin1* overexpression restored the effects of GV1001 (Figure 3E). Those results indicate that GV1001 regulates osteoblast differentiation in a Pin1-dependent manner.

### 3.4. GV1001 Regulates Pin1-Mediated Osteoblast Differentiation

To further validate the above results, the Pin1 overexpression plasmid was used to follow the experiments. The overexpression of *Pin1* markedly enhanced the GV1001-induced ALP activity and mineralization in C2C12 cells (Figure 4A). Likewise, the overexpression of *Pin1* increased the mRNA expression and protein levels of genes associated with osteogenesis (Figure 4B,C). The luciferase reporter gene assay verified that GV1001-induced transcriptional activity was further increased by *Pin1* overexpression (Figure 4D). As OCN synthesized by osteoblasts is a specific marker for osteogenic maturation [24], the effects of GV1001 and Pin1 on the terminal differentiation of osteoblasts were also assessed by determining the production of OCN. As shown in Figure 4E, treatment with GV1001 increased the levels of OCN. Subsequently, *Pin1* overexpression further enhanced the GV1001-induced OCN production. Those results show that Pin1 is involved in GV1001-regulated osteoblast differentiation.

### 3.5. GV1001 Enhances the Osteoblast Differentiation via Stabilization of the Runx2 Protein

The interaction between Pin1 and Runx2 during osteoblast differentiation is characterized by the Pin1-mediated stabilization of phosphorylated Runx2, enhancing its transcriptional activity and contributing to the regulation of genes involved in osteogenesis and bone formation [25]. To identify the mechanisms by which GV1001 promotes osteoblast differentiation, the relevance of Runx2 was investigated. Consistent with the results shown in Figure 2, the activity of ALP and the formation of calcium nodules, as demonstrated by ARS staining, were increased upon stimulation with BMP4 and were further increased after the overexpression of *Runx2*. In addition, the increase in osteoblast differentiation mediated by Runx2 was significantly enhanced by GV1001 treatment (Figure 5A). These results were similar to the mRNA expression of *Alp*, *Bsp*, and *Col1a1*, which are related to osteoblast differentiation (Figure 5B). Our previous results showed that GV1001 increased the protein abundance of endogenous Runx2 (Figure 2C). Runx2 was exogenously overexpressed to determine the effect of GV1001 on the corresponding protein abundance. GV1001 increased the protein abundance of Runx2 in a dose-dependent manner (Figure 5C). Next, the half-life of Runx2 was observed with or without GV1001 by treatment with CHX, a protein synthesis inhibitor. The presence of GV1001 increased the half-life of Runx2 by approximately 1.5 h to 3.5 h (Figure 5D). The effect on transcription activity was also examined. The transcription activity was increased by Runx2 and was further increased by treatment with GV1001 in both ALP and BSP promoters (Figure 5E). Those results show that GV1001 enhances osteoblast differentiation via enhancing Runx2 stability and transcriptional activity.

### 3.6. GV1001 Also Increases the Osteoblast Differentiation by Stabilization of Osterix Protein

Osterix, along with Runx2, is known as the most important transcription factor mediating osteoblast differentiation, and Pin1 also influences Osterix during osteoblast differentiation by catalyzing the isomerization of phosphorylated proline-directed serine or threonine residues, enhancing Osterix stability and transcriptional activity, thereby promoting osteoblast differentiation and the expression of genes involved in bone formation [26]. The following experiment was conducted to determine the effect of GV1001 on Osterix. Upon stimulation with BMP4, *Osterix* overexpression increased ALP activity and calcium deposition (Figure 6A). Furthermore, with respect to the mRNA expression of the osteogenic markers, the *Alp* level was increased similarly by the overall concentration of GV1001, and the levels of *Bsp* and *Col1a1* were enhanced by GV1001 in a dose-dependent manner (Figure 6B). In addition, GV1001 increased the protein abundance of overexpressed Osterix compared to that observed in the non-treated control group at concentrations above 0.2 µg/mL (Figure 6C). Half-life verification experiments with CHX showed that the half-life of Osterix remained almost constant in the presence of GV1001 (Figure 6D). The transcription activity was measured using ALP and BSP promoters. The transcription activity was increased by Osterix and was increased further after treatment with GV1001 (Figure 6E). Those results show that GV1001 not only regulates Runx2 function but also increases Osterix stability and transcriptional activity during osteogenesis.

### 3.7. GV1001 Reduces OVX-Induced Bone Destruction in Mice

To determine whether GV1001 could recover bone loss, mice were either sham operated or ovariectomized with or without GV1001 treatment. Four weeks after surgery, ovariectomy caused a marked decrease in BMD in mice. However, treatment with GV1001 for 3 weeks resulted in a significant increase in BMD compared to that observed in OVX mice (Figure 7A). Consistently, morphometric µCT analysis of trabecular bone also revealed that bone loss in OVX mice was recovered by GV1001 (Figure 7B). Next, various parameters (BV/TV, Tb.Th, Tb. N, SMI, Tb.Pf, and Tb. Sp) were calculated from the µCT data for each mouse. The values of BV/TV, Tb.N, and Tb.Th in OVX mice were lower than those observed in the SHAM group. However, the changes in these parameters were significantly restored in GV1001-treated mice (Figure 7C). In contrast, the values of SMI, Tb.Pf, and Tb.Sp of the femur were increased in the OVX group compared to those observed in the SHAM group. The administration of GV1001 repressed this effect in OVX mice (Figure 7C). Those results support that GV1001 has an important role in bone formation.

## 4. Discussion

Many studies have been conducted to examine the factors promoting osteoblast differentiation to improve osteoporosis treatments [27]. Although parathyroid hormone agents are typically used in clinical trials, they have limitations in clinical use owing to different medicinal effects followed by changes in the concentration and lower efficacy compared to bisphosphonates, the typical osteoporosis drugs [28]. Therefore, various efforts are still being undertaken to enhance osteoblast differentiation using different treatment methods. The present study was conducted using peptides that are considered relatively safe substances. So far, many naturally occurring peptides have been identified, which act as physiological substances, such as hormones, neurotransmitters, and growth factors in the body [29]. In general, peptides are the starting point of intracellular signaling pathways along with their receptors. Hence, they demonstrate important intrinsic functions and pharmacological effects and exhibit the potential to be developed as a new treatment [10]. In terms of stability, tolerability, and efficacy, the potency can be distinguished from the existing small molecules. In addition, peptide therapy has the advantage of low production complexity, unlike other protein-based medicines, and demonstrates similar production access as small molecules [30]. Currently, the list of diseases that are primarily treated using peptide drugs is led by diabetes, a metabolic disease, which is also observed in association with various cancers and obesity. The representative peptide drugs are glucagon-like peptide-1 (GLP-1) agonists used in the treatment of type 2 diabetes and somatostatin receptor agonists used in the treatment of Cushing’s syndrome [31,32]. Currently, many peptides are being investigated in clinical trials for the treatment of conditions, such as inflammation and infectious diseases. Therefore, the study of peptides aimed at treating osteoporosis is significant.

Biotin–streptavidin-based protein screening offers a highly specific and versatile method for accurately identifying protein interactions across diverse biological contexts, facilitating comprehensive insights into cellular processes and potential therapeutic targets. GV1001, when conjugated with biotin, was synthesized to investigate its interactions with proteins using affinity-based techniques (Figure 1B,C). Through proteomic analysis, GV1001 was found to strongly bind to PPIA, which plays a crucial role in bone metabolism and osteogenesis regulation via its modulation of Pin1 (Figure 1E). Co-IP confirmed the direct interaction between GV1001 and Pin1 (Figure 1F). The direct binding to Pin1 is crucial for understanding the therapeutic implications of osteoporosis. While we have partially confirmed this interaction through Co-IP, we have further evaluated this direct binding using molecular docking analysis. Our docking results provide additional evidence supporting the interaction between GV1001 and Pin1 (Appendix A), strengthening our understanding of their binding relationship. Furthermore, the Green Pin1 activity assay revealed that GV1001 significantly enhanced Pin1 catalytic activity, suggesting its potential role in modulating Pin1 function. Conversely, treatment with tannic acid, known to inhibit Pin1 activity, led to a suppression of Pin1 activity (Figure 1H). These findings suggest that GV1001 regulates Pin1 activity through direct interaction, highlighting its potential therapeutic implications in osteoporosis involving Pin1 dysregulation. 

C2C12 pre-myoblast cells, known to differentiate into osteoblasts in a differentiation medium containing BMP-4, are suitable for validating the osteogenic activity of drugs or targets [33]. Especially, in response to the BMP signal, the default myoblast differentiation will be blocked and the C2C12 cells will differentiate toward the osteoblast lineage [34,35]. In the present study, GV1001 effectively increased the activity of ALP, a marker of early osteoblast differentiation and calcium formation, thereby improving bone mineralization and promoting osteoblast differentiation upon the stimulation of BMP-4 in the C2C12 cell line (Figure 2). In particular, GV1001 stabilized the protein levels of Runx2 and Osterix, resulting in the increased overall activity of the two transcription factors (Figure 5D and Figure 6D). The mRNA expression of the striata-related markers was increased along with the corresponding transcription activity, eventually resulting in differentiation. In the process of osteoblast differentiation, the two transcription factors are known to work at different stages [36]. Runx2 is known to be expressed early in the differentiation, while Osterix is a relatively later transcription factor primarily acting on embryos for 13.5 days during mouse embryonic development [37]. A time difference is observed between the activity of the two transcription factors, which are the factors involved in the maturing of the osteoblast rather than the initial differentiation of the osteoblast [38]. Surprisingly, GV1001 demonstrated both the stabilization and activation of Runx2 and Osterix, which exert different effects. These results show that the effects of GV1001 could potentially influence the entire osteoblast differentiation process. Particularly, controlling the activity of transcription factors leads to protein stabilization. Previous studies have reported that the protein stabilization of Runx2 and Osterix is caused by various methods, including phosphorylation, methylation, and other modifications [39,40]. Although the upstream target genes that control both Runx2 and Osterix at the same time are not very well known, Pin1, a peptidyl-prolyl cis-trans isomerase, is reported to regulate both transcription factors at the protein level via a direct interaction [26,41]. In our study, the protein abundance of Runx2 and Osterix was decreased in the absence of Pin1, and subsequently, GV1001-induced activity was entirely attenuated (Figure 3). Meanwhile, Pin1 overexpression markedly enhanced GV1001-induced osteogenic activity (Figure 4). These results suggest that GV1001 controls the activity and stability of Runx2 and Osterix by regulating the upstream gene, Pin1, and ultimately affects osteoblast differentiation. The osteogenic activity of GV1001 was also observed in the μCT scanning results obtained after in vivo experiments. OVX mice treated for 3 weeks with GV1001 demonstrated improved trabecular femur BMD and other bone morphometric parameters (BV/TV, Tb.N, and Tb.Th) that are associated with bone loss after OVX (Figure 7). 

GV1001 has been investigated in a variety of in vivo studies to assess its potential therapeutic effects in various disease models, such as cancer and inflammatory disease [42,43,44]. The dosing regimen used in our study (0.01–10 mg/kg, five times per week for 3 weeks) was chosen based on our preliminary experiments and a review of the literature. Our preliminary experiments indicated that this dose was effective in promoting bone formation without causing any apparent toxicity. We did not observe any adverse effects associated with GV1001 administration. This is consistent with previous findings in in vivo models, which have not reported any significant toxicity or safety concerns related to GV1001 treatment [42,43,44]. The overall safety profile of GV1001 is further supported by its lack of negative effects on body weight or general behavior in our study. In the end, the optimization of the GV1001 dosing regimen is crucial for maximizing its therapeutic potential while minimizing any potential side effects. 

## 5. Conclusions

In conclusion, our data demonstrate that GV1001 upregulates BMP4-induced osteoblast differentiation in C2C12 cells by enhancing the activity and stability of Runx2 and Osterix in a Pin1-dependent manner. The present study also demonstrated the beneficial effect of GV1001 on OVX-induced bone resorption. Therefore, GV1001 could be proposed as an alternative to a novel treatment and could compensate for the side effects of current osteoporosis treatments.

## Figures and Tables

**Figure 1 biomolecules-14-00812-f001:**
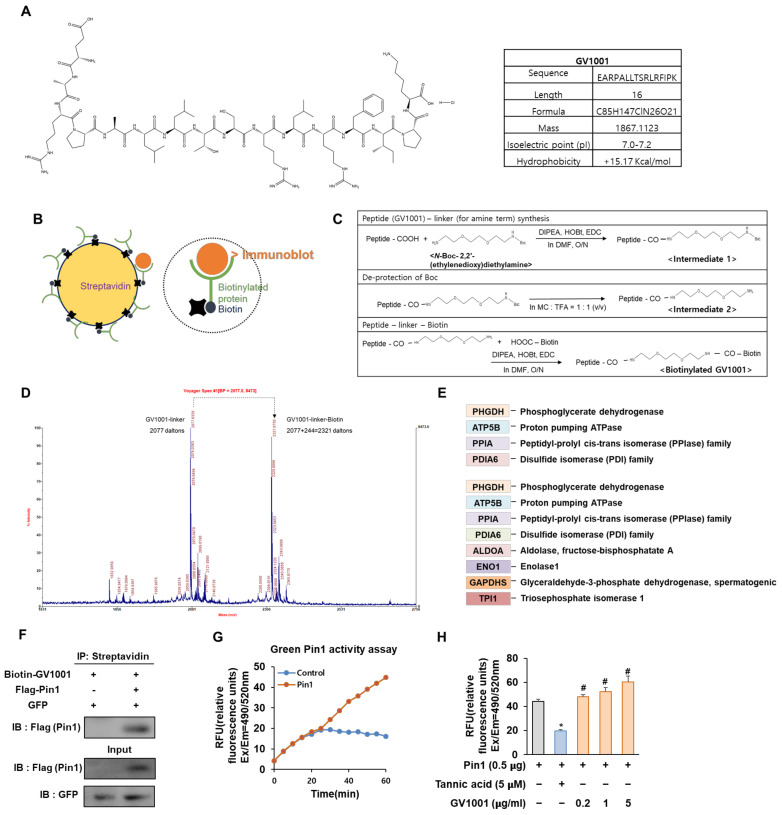
Biotinylated GV1001 interacts with peptidyl-prolyl isomerase A (PPIA), regulating Pin1 activity directly. (**A**) Chemical structure and physical and chemical properties of GV1001. (**B**) The schematic structure of streptavidin (**left**). The schematic structure of binding between streptavidin-biotin (**right**). (**C**) Biotin conjugation reaction to GV1001. Step 1 reagents and conditions: DIPEA, HOBt, EDC, DMF, overnight; step 2 reagents and conditions: MC, TFA (quantitative); step 3 reagents and conditions: DIPEA, HOBt, EDC, DMF, overnight. (**D**) Mass spectrometry to detect the biotinylation of GV1001. Two distinct peaks are indicated as a GV1001 linker (2077 Da) and a GV1001 linker—biotin (2077 + 244 = 2321 Da). (**E**) Protein list generated via identification analysis screens for GV1001 peptide binders, detecting significant differences between streptavidin and biotinylated samples. (**F**) The interaction between Pin1 and GV1001. HEK 293 cells were co-transfected with combinations of the FLAG-empty vector or FLAG-Pin1 along with the GFP-empty vector as a transfection control. The protein interaction between GV1001 and Pin1 was confirmed by co-IP using streptavidin followed by IB for Pin1 using a FLAG antibody IB: FLAG (Pin1). The levels of overexpressed GFP and FLAG-Pin1 in cell lysates are also examined. (**G**) Catalyzed activity of Pin1 measured with a Green Pin1 activity assay kit. The increase in fluorescence intensity is directly proportional to Pin1 activity. Fluorescence is monitored at Ex/Em = 490/520 nm. (**H**) Measurement of Pin1 activity modulation by GV1001. Tannic acid (Pin1 inhibitor, 5 µM) or GV1001 (0.2, 1, or 5 µg/mL) were incubated with a Pin1 enzyme for 1 h at room temperature. The substrate was subsequently added to the reaction for another 30 min of incubation. The final reaction was then initiated by adding the developer to each well and detected at Ex/Em = 490/520 nm. * *p* < 0.05 compared with the Pin1-transfected control group; # *p* < 0.05 compared with the Pin1-transfected and tannic acid-treated group. The experiment was performed in triplicates, and the average and SDs are shown.

**Figure 2 biomolecules-14-00812-f002:**
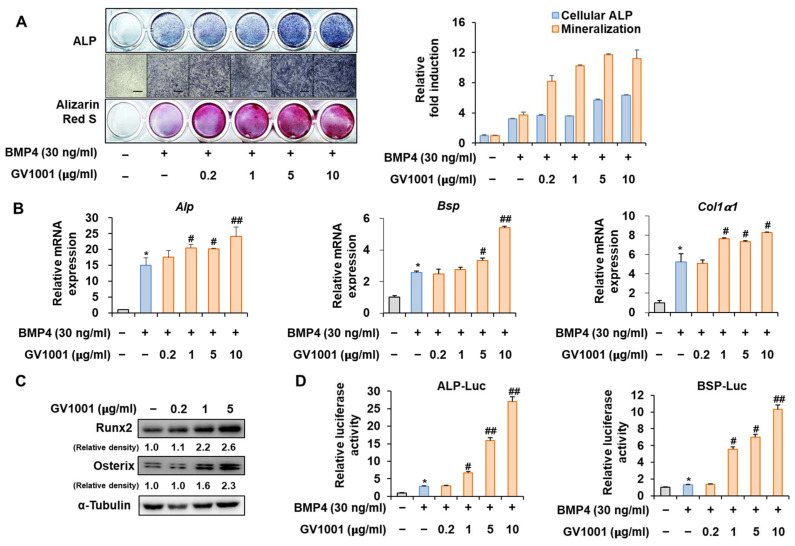
Stimulation of osteoblast differentiation by GV1001 in C2C12 cells. C2C12 cells were treated with GV1001 (0.2, 1, 5, or 10 µg/mL) after stimulation with BMP4 (30 ng/mL). (**A**) ALP staining and ARS staining of C2C12 cells after 3 or 7 days, respectively. The ratio of relative cellular ALP and mineralization was normalized to that of the control group. The positive cells showed purple nuclear and blue-purple granules in the cytoplasm. The calcium deposits showed red after ARS staining. (scale bar = 100 μm). (**B**) After 7 days, the mRNA expression levels of the osteoblast-specific markers, Alp, Bsp, Col1α1, and Runx2, were determined using real-time PCR and normalized to Gapdh. * *p* < 0.05 compared with the control group; # *p* < 0.05 and ## *p* < 0.01 compared with the BMP4-treated group. (**C**) The protein levels of Runx2 and Osterix were examined using IB. α-Tubulin was used as a loading control. The intensities of protein bands were analyzed using Multi Gauge and indicated by numbers at the bottom of the corresponding bands after normalized to the values of α-Tubulin. (**D**) C2C12 cells were co-transfected with plasmids containing luciferase reporters (ALP-Luc or BSP-Luc) along with pCMV-β-gal as a reference control. The cells were treated with GV1001 (0.2, 1, 5, or 10 µg/mL) after stimulation with BMP4. After 24 h, luciferase activities were measured. * *p* < 0.05 compared with the control group; # *p* < 0.05 and ## *p* < 0.01 compared with the BMP4-treated group. The experiment was performed in triplicates, and the average and SDs are shown.

**Figure 3 biomolecules-14-00812-f003:**
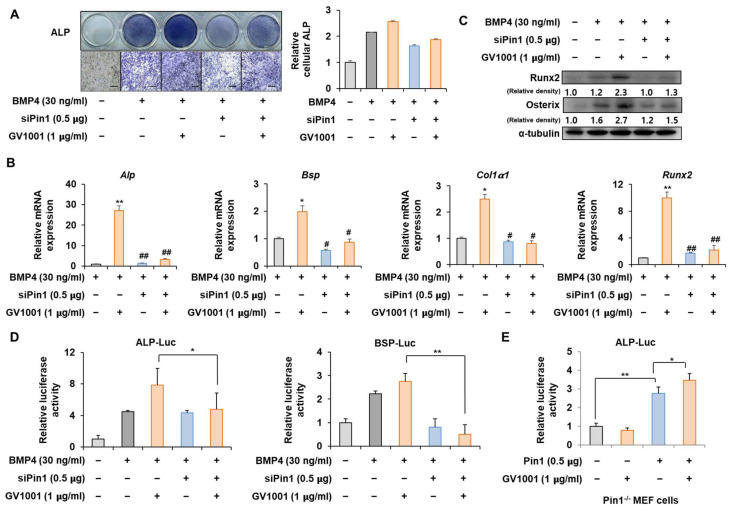
Effect of Pin1 knockdown on GV1001-induced osteoblast differentiation. C2C12 cells were transfected with pSuper-Pin1 (shPin1) and then treated with BMP4 (30 ng/mL) and GV1001 (1 µg/mL). (**A**) The extent of ALP activity was evaluated using ALP staining for 3 days. The ratio of relative cellular ALP was normalized to that of the control group. The positive cells showed purple nuclear and blue-purple granules in the cytoplasm (scale bar = 100 μm). (**B**) After 7 days, the mRNA expression levels, *Alp*, *Bsp*, *Col1α1*, and *Runx2,* were determined using real-time PCR and normalized to Gapdh. * *p* < 0.05 and ** *p* < 0.01 compared with the control group; # *p* < 0.05 and ## *p* < 0.01 compared with the BMP4-treated group. (**C**) The protein abundance of Runx2 and Osterix was analyzed using IB. α-Tubulin was used as a loading control. The intensities of protein bands were analyzed using Multi Gauge V3.0 (FUJIFILM) image software and indicated by numbers at the bottom of the corresponding bands after being normalized to the values of α-Tubulin. (**D**) C2C12 cells were transfected with a luciferase reporter (ALP-Luc, BSP-Luc; 0.3 μg), pCMV-β-gal (0.1 μg), and shPin1 (0.5 μg) and exposed to GV1001 (1 µg/mL). (**E**) Pin1 knockout and Pin1-/- mouse embryonic fibroblast (MEF) cells were transfected with a luciferase reporter (ALP-Luc; 0.3 μg), pCMV-β-gal (0.1 μg), and Pin1 (0.5 μg) and exposed to GV1001 (1 µg/mL). The luciferase activities were measured after 24 h. * *p* < 0.05 and ** *p* < 0.01. The experiment was performed in triplicates, and the average and SDs are shown.

**Figure 4 biomolecules-14-00812-f004:**
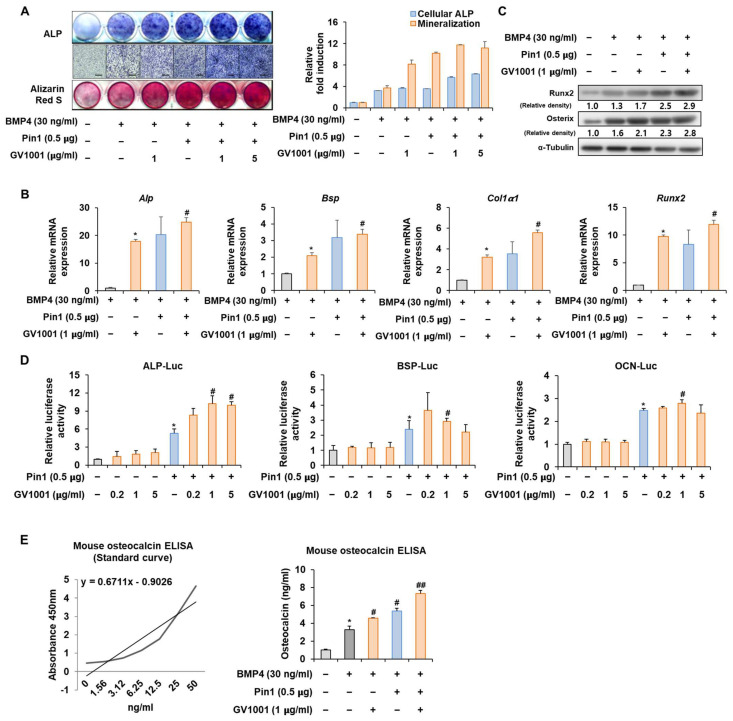
Effect of GV1001 on *Pin1* overexpression during osteoblast differentiation. C2C12 cells were transfected with Pin1 (0.5 μg) and then treated with BMP4 (30 ng/mL) and GV1001 (1 or 5 µg/mL for panel (**A**), 1 μg/mL for panels (**B**,**C**)). (**A**) ALP staining and ARS staining of C2C12 cells after 3 or 7 days, respectively. The ratio of relative cellular ALP and mineralization was normalized to that of the control group. The ALP-positive cells showed purple nuclear and blue-purple granules in the cytoplasm. The calcium deposits showed red after ARS staining. (**B**) After 7 days, the mRNA expression levels of the osteoblast-specific markers, *Alp*, *Bsp*, *Col1α1*, and *Runx2*, were determined using real-time PCR and normalized to *Gapdh*. * *p* < 0.05 compared with the BMP4-treated group; # *p* < 0.05 compared with the GV1001-induced group. (**C**) The protein abundance of Runx2 and Osterix was analyzed using IB. α-Tubulin was used as a loading control. The intensities of protein bands were analyzed using Multi Gauge V3.0 (FUJIFILM) image software and indicated by numbers at the bottom of the corresponding bands after being normalized to the values of α-Tubulin. (**D**) C2C12 cells were transfected with a luciferase reporter (ALP-Luc, BSP-Luc, or OCN-Luc; 0.3 μg), pCMV-β-gal (0.1 μg), and Pin1 (0.5 μg) and exposed to GV1001 (0.2, 1, or 5 µg/mL). The luciferase activities were measured after 24 h. * *p* < 0.05 compared with the control group; # *p* < 0.05 compared with the Pin1-transfected group. (**E**) C2C12 cells were transfected with Pin1 (0.5 μg) and then treated with BMP4 (30 ng/mL) and GV1001 (1 μg/mL). After 7 days, the amount of OCN in the culture medium was assessed using an osteocalcin ELISA kit. * *p* < 0.05 compared with the BMP4-untreated group; # *p* < 0.05 and ## *p* < 0.01 compared with the BMP4-treated group. The experiment was performed in triplicates, and the average and SDs are shown.

**Figure 5 biomolecules-14-00812-f005:**
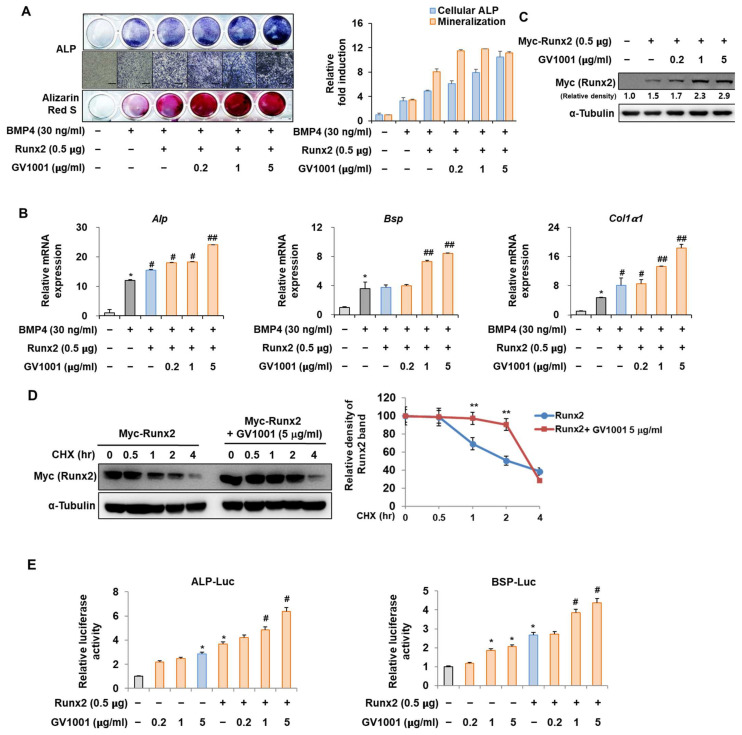
Regulation of Runx2-induced osteogenic activity by GV1001. C2C12 cells were transfected with Myc-Runx2 (0.5 μg) and then treated with BMP4 (30 ng/mL) and GV1001 (0.2, 1, or 5 µg/mL). (**A**) ALP staining and ARS staining of C2C12 cells after 3 or 7 days, respectively. The ratio of relative cellular ALP and mineralization was normalized to that of the control group. The ALP-positive cells showed purple nuclear and blue-purple granules in the cytoplasm. The calcium deposits showed red after ARS staining (scale bar = 100 μm). (**B**) After 7 days, the mRNA expression levels of the osteoblast-specific markers, *Alp*, *Bsp*, and *Col1α1,* were determined using real-time PCR and normalized to *Gapdh*. * *p* < 0.05 compared with the control group; # *p* < 0.05 and ## *p* < 0.01 compared with the BMP4-treated group. (**C**) C2C12 cells were transfected with Myc-Runx2 (0.5 μg) and then treated with GV1001 (0.2, 1, or 5 µg/mL) for 3 days. The cell lysates were subjected to IB. The intensities of protein bands were analyzed using Multi Gauge and indicated by numbers at the bottom of the corresponding bands after normalized to the values of α-Tubulin. (**D**) C2C12 cells were transfected with Myc-Runx2 and treated with GV1001 (5 µg/mL). After 3 days, the cells were treated with cycloheximide (CHX; 40 μg/mL) to determine the half-life of Runx2 at the indicated time points. The cell lysates were subjected to IB. The intensities of the Myc-Runx2 bands were analyzed using Multi Gauge. The protein abundance of Runx2 in CHX-untreated cells (0 h) was set to 100%. ** *p* < 0.01 compared with the control group. (**E**) C2C12 cells were transfected with a luciferase reporter (ALP-Luc or BSP-Luc; 0.3 μg), pCMV-β-gal (0.1 μg), and Runx2 (0.5 μg) and exposed to GV1001 (0.2, 1, or 5 µg/mL). The luciferase activities were measured after 24 h. * *p* < 0.05 compared with the control group; # *p* < 0.05 compared with the Runx2-transfected group. The experiment was performed in triplicates, and the average and SDs are shown.

**Figure 6 biomolecules-14-00812-f006:**
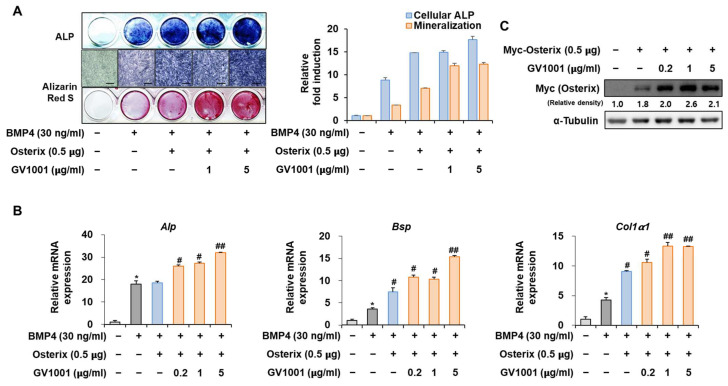
Regulation of Osterix-induced osteogenic activity by GV1001. C2C12 cells were transfected with Osterix (0.5 μg) and then treated with BMP4 (30 ng/mL) and GV1001 (1 or 5 µg/mL). (**A**) ALP staining and ARS staining of C2C12 cells after 3 or 7 days, respectively. The ratio of relative cellular ALP and mineralization was normalized to that of the control group. The ALP-positive cells showed purple nuclear and blue-purple granules in the cytoplasm. The calcium deposits showed red after ARS staining (scale bar = 100 μm). (**B**) After 7 days, the mRNA expression levels of the osteoblast-specific markers, *Alp*, *Bsp*, and *Col1α1,* were determined using real-time PCR and normalized to *Gapdh*. * *p* < 0.05 compared with the control group; # *p* < 0.05 and ## *p* < 0.01 compared with the BMP4-treated group. (**C**) C2C12 cells were transfected with Myc-Osterix (0.5 μg) and then treated with GV1001 (0.2, 1, or 5 µg/mL) for 3 days. The cell lysates were subjected to IB. The intensities of protein bands were analyzed using Multi Gauge and indicated by numbers at the bottom of corresponding bands after normalized to the values of α-Tubulin. (**D**) C2C12 cells were transfected with Myc-Osterix and treated with GV1001 (5 µg/mL). After 3 days, the cells were treated with CHX (40 μg/mL) for indicated time points, and the cell lysates were subjected to IB. The intensities of Myc-Osterix bands were analyzed using Multi Gauge. The protein abundance of Osterix in CHX-untreated cells (0 h) was set to 100%. ** *p* < 0.01 compared with the control group (**E**) C2C12 cells were transfected with a luciferase reporter (ALP-Luc or BSP-Luc; 0.3 μg), pCMV-β-gal (0.1 μg), and Osterix (0.5 μg) and exposed to GV1001 (0.2, 1, or 5 µg/mL). The luciferase activities were measured after 24 h. * *p* < 0.05 compared with the control group; # *p* < 0.05 compared with the Osterix-transfected group. The experiment was performed in triplicates, and the average and SDs are shown.

**Figure 7 biomolecules-14-00812-f007:**
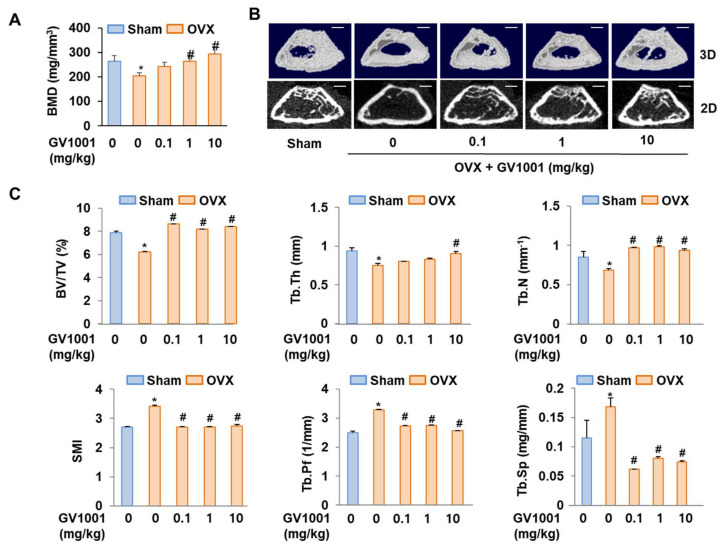
Preventive effects of GV1001 on bone loss in OVX mice. GV1001 (0.1, 1, or 10 mg/kg) was administered to SHAM and OVX mice via a subcutaneous injection 5 days per week for 21 days. (**A**) Bone marrow density (BMD) was measured using DEXA. (**B**) The representative 2D and 3D trabecular tomographic images of seven femurs were analyzed using µCT (scale bar = 1 mm). (**C**) Bone morphometric parameters, including bone volume/tissue volume (BV/TV), trabecular thickness (Tb.Th), trabecular number (Tb.N), structure model index (SMI), and trabecular pattern factor (Tb.Pf), were determined. * *p* < 0.05 compared with the SHAM group; # *p* < 0.05 compared with the OVX group.

**Table 1 biomolecules-14-00812-t001:** Primer sequences used for RT-PCR.

Gene	Primer Sequence (5′→3′)
*Alp*	(F) 5′-GGA CAT GCA GTA CGA GCT GA-3′
	(R) 5′-GCA GTG AAG GGC TTC TTG TC-3′
*Bsp*	(F) 5′-GCG AAG CAG AAG TGG ATG AAA -3′
	(R) 5′-TGC CTC TGT GCT GTT GGT ACT G -3′
*Col1α1*	(F) 5′-CTG ACC TTC CTG CGC CTG ATG TCC-3′
	(R) 5′-GTC TGG GGC ACC AAC GTC CAA GGG-3′
*Runx2*	(F) 5′-AGC AAC AGC AAC AGC AG-3′
	(R) 5′-GTA ATC TGA CTC TGT CCT TG-3′
*Gapdh*	(F) 5′-ACC ACA GTC CAT GCC ATC A-3′
	(R) 5′-TCC ACC ACC CTG TTG CTG T--3′

## Data Availability

Please contact the corresponding author for reasonable data requests.

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
