# Peer review of "The Cell-Penetrating Peptide GV1001 Enhances Bone Formation via Pin1-Mediated Augmentation of Runx2 and Osterix Stability"

_biomolecules, 2024, doi:10.3390/biom14070812_

Round 1
Reviewer 1 Report
Comments and Suggestions for Authors
This manuscript entitled “The cell-penetrating peptide GV1001 enhances bone formation via Pin1-mediated augmentation of Runx2 and Osterix stability” found that GV1001 demonstrated protective effects against bone loss in OVX mice by upregulating osteogenic differentiation via Pin1-mediated protein stabilization of Runx2 and Osterix. GV1001 could be a potential candidate with anabolic effects for the prevention and treatment of osteoporosis. However, several issues need to be clarified before publication.
1 There are multiple amines and carboxylic acids on GV1001. How to avoid unwanted coupling of biotin on its side chains or N-terminal during the biotinylated reaction?
2 The article mentions that biotinylated GV1001 binds to PPIA, particularly Pin1. The direct binding to Pin1 essentially executes its therapeutic implications on OP. More proofs are recommended to evaluate this direct binding, for example, CD evaluation and docking.
3 In vivo experiments are performed to demonstrate the potential therapeutic potentials for treating osteoporosis, however, the assessments on the stability and potential toxicity of GV1001 are lacking.
4 In the introduction the authors state that GV1001 is not positively charged at physiological pH (page 2 line 70). Figure 1A shows the pI at 12.20 and a net charge of +3. They are self-contradictory. Please check.
5 Please check and confirm the specificity of the primer sequences used for RT-PCR.
6 All the staining images are fuzzy. Please provide high-resolution images.
7 The significance statement (* and #) in Figure 1H is not included.
Comments on the Quality of English Languageno comments
Reviewer 2 Report
Comments and Suggestions for Authors
The manuscript “The cell-penetrating peptide GV1001 enhances bone formation 2 via Pin1-mediated augmentation of Runx2 and Osterix stability” prepared by Piao, M. et al demonstrates the use of the cell penetrating peptide GV1001 to enhance bone formation both in vitro and in vivo. The authors show that GV1001 binds to Pin1, a PPIA, which subsequently stablilizes Runx2 and Osterix, 2 essential genes promoting bone differentiation and maturation, leading to enhanced bone formation when compared to the non-GV1001 treated counterpart. Although the data look promising, the authors need to address several issues before the manuscript is ready for publication.
Line 47: should it be “does NOT meet”?
Line 75-80: In the last paragraph of introduction, the authors should rewrite or shuffle the sentences. Typically, the last paragraph describes what work has been done, but the start of this paragraph with the description of the biotin-streptavidin system deviates the main point. The authors should first describe the construction of the biotinylated GV1001 first before describing why biotinylated protein was generated (i.e. how the biotin-streptavidin help to address the binding partner(s) of GV1001.
Line 136: What do the authors mean by “was combined indicated combination”?
Line 136-137: “was allowed to incubate”. Under what conditions?
Line 179-180: The authors mentioned that they used β-galactosidase (β-gal) as a control to analyze the transfection efficiency. How did they do it? Counting the number of β-gal positive cells and comparing to the total number of cells? By manually counting? Or using some software for quantification?
Line 187: analysis
Line 187: 100 μL of cell culture medium that was collected at different points during osteogenic differentiation were used.
Line 200: Why was GV1001 only given 5 days per week but not daily? Is there any side effect of GV1001 with continuous administrations? The authors should discuss adverse effect of GV1001 somewhere in the manuscript and mention any side effect they have observed when using GV1001?
Line 216-219: Why was only ANOVA used for statistical analysis? ANOVA compares results from 2 or more groups. For comparison of any treatment group to the control group, pairwise tests such as Student’s t-test should also be performed.
Line 245: Define (Given full name of ) PPIA
Line 248: Define (Given full name of ) Pin1
Figure 1C: Seems a bond missing between CO and HN in Intermediate 1 in the first reaction scheme.
Line 296-298: What do the authors mean by Col1a1 was more pronounced BMP4 stimulation in Figure 2B? No BMP4 stimulation is definitely showing lower Col1a1 according to the graph.
Section 3.3 and 3.4: It looks more logical to first knockdown a gene and evaluate its contribution before overexpression.
Line 325-327: The authors claimed that overexpression of Pin1 was enhanced by GV1001. In this experiment, overexpression of Pin1 was driven by the CMV promoter. Is GV1001 enhancing Pin1 expression by increasing promoter/transcription activity or by stabilizing the protein? More important will be how GV1001 affects the endogenous Pin1 gene expression and protein stability. The authors should check if gene expression (PCR data) or protein stability (Western Blot) of Pin1 changes with various concentrations of GV1001. This study is equally important as the stability study of Runx2 and Osterix presented in order to uncover the mechanism behind GV1001 mediated osteogenesis.
Line 362-363: There is no Pin1 overexpression in Figure 4E. The label shows shPin1. Is this a wrong label?
Line 367: Figure 4 legend title should either be “Effect of GV1001 on Pin1 knockdown cells during osteoblast differentiation” or “Effect of Pin1 knockdown on GV1001-induced osteoblast differentiation.
Line 394: Figure 2D does show any Runx2.
Line 398: Cycloheximide (CHX) should be protein synthesis inhibitor not proteasomal inhibitor.
Line 477: These results….
Line 527-528: It is better to mention that BMP4 promotes osteogenesis in C2C12 cells in the Results section to avoid confusion.
Line 543-544: While the authors claim that GV1001 does not act at one point during osteogenesis, the data presented in the manuscript does not support this claim. What the shows is that GV1001 act at one point (Pin1), which subsequently has multiple effects such as Runx2 and Osterix stability and other downstream transcriptional activities. Unless the authors also analyze the other binding partners of GV1001 showed in Figure 1, the data presented in this manuscript only point toward one target (Pin1).
Comments on the Quality of English Language
/
Round 2
Reviewer 1 Report
Comments and Suggestions for Authors
The authors have addressed all my concerns. The revised manuscript merits publication in Biomolecules now.